# Shapeshifters: Auditory cortical neurons switch from polysemantic to monosemantic under anesthesia

## Abstract

General anesthesia transitions the brain from a conscious to an unconscious state, but how does sensory processing differ between these conditions? To address this question, we trained neural network encoding models to predict the responses of auditory cortical neurons to natural sounds in both awake and anesthetized ferrets. Utilizing mechanistic interpretability methods, such as feature visualization, linearization and sparse autoencoders, we analyzed these networks tuning and connectivity to uncover key differences in sensory processing. We found that anesthesia decouples neural connectivity, shifting neurons from polysemantic (responding to multiple inputs) to monosemantic (responding to a single input), resulting in a lower-dimensional population code. These findings illuminate how anesthesia alters neural connectivity and encoding, offering new insights into the neural mechanisms underlying sensory processing.

## 1 Introduction

General anesthetics have the remarkable ability to shift a conscious mind into an unconscious state, marking a significant milestone in modern medicine by alleviating patient suffering during major surgery (Voss et al., 2019). Much of our understanding of visual and auditory processing in the brain has been derived from studies on anesthetized animals, based on the assumption that sensory processing is largely unaffected by anesthesia (Gaese & Ostwald, 2001; Sellers et al., 2015). However, this assumption is now known to be inaccurate, as anesthesia has been shown to alter sensory processes (Fontanini & Katz, 2008). In the auditory cortex, anesthesia modulates the variability of auditory-evoked responses (Kisley & Gerstein, 1999), reduces the number of active neurons, increases response onset latency (Cheung et al., 2001; Noda & Takahashi, 2015), and inhibits sustained firing to preferred sounds (Wang et al., 2005). Despite these findings, it remains largely unclear how the spectrotemporal tuning properties of auditory cortical neurons are affected by anesthesia.

In this work, we used neural networks to predict the responses of neurons in the primary auditory cortex (A1) of awake and anesthetized ferrets to natural sounds. Using advances in machine learning interpretability techniques, we analyzed differences in tuning and connectivity between these networks to reveal key distinctions in auditory cortical processing between awake and anesthetized states. Our contributions are summarized as follows:

1. We found that adding a gated output to various encoding models of the auditory cortex improves their performance and obtains state-of-the-art response prediction.

2. We demonstrated how the highly nonlinear spectrotemporal tuning of awake A1 neurons can be visualized using feature visualization (Olah et al., 2017).

3. We identified awake A1 neurons as polysemantic (tuned to different spectrotemporal inputs) and anesthetized A1 neurons as monosemantic (tuned to fixed spectrotemporal inputs) using an ANN-linerization technique (Keshishian et al., 2020).

4. Using principal component analysis (PCA) and sparse autoencoders (SAEs) (Olshausen & Field, 1997), we showed that the population code of awake A1 neurons has a higher dimensionality compared to that of anesthetized neurons.

5. Finally, we found that the neural network trained on anesthetized responses exhibits sparser connectivity to the output units, suggesting that anesthesia decouples cortical connectivity.

## 2 METHODS

**A1 neural datasets**  We utilized two publicly available datasets of single-unit responses to natural sounds recorded in the primary auditory cortex (A1) (and anterior auditory field) of awake and anesthetized ferrets. The awake dataset contains responses from $N = 185$ neurons to 299 4-second sound clips (Lopez Espejo et al., 2019), while the anesthetized dataset consists of responses from $N = 73$ neurons to 20 5-second sound clips (Harper et al., 2016). Anesthesia was induced using a combination of ketamine and medetomidine. We excluded four sound clips from training in both datasets and reserved them for testing model performance. The anesthetized dataset has 20 repeats for each sound in the training and test set, and the awake dataset has no repeats for the training set and 10 repeats for the test set. Neuronal responses were binned at 4ms intervals and we trained the encoding models to predict the trial-averaged responses to the sounds. Both datasets include natural sounds of human speech, animal vocalizations, and environmental sounds, and exhibit a similar average power spectrum, with power declining from the higher to the lower frequencies (Appendix Fig. 1), as has similarly been reported in prior work (Machens et al., 2004). Both datasets exhibit similar firing rate distributions, with the firing rates being slightly more elevated for the awake (14.28Hz) than the anesthetized (12.54Hz) responses (Appendix Fig. 2).

**Cochleagrams**  Encoding models of the auditory cortex typically use spectrograms (or cochleagrams) - frequency-decomposed representations approximating cochlear transformations - rather than raw sound waveforms (Linden et al., 2003; Gill et al., 2006; David et al., 2009; Harper et al., 2016). We constructed each sound clip's cochleagram by applying a short-time Fourier transform with a 4ms temporal resolution; calculating the power over $C = 64$ logarithmically spaced frequency channels (500-22000Hz) via weighted summation of overlapping triangular windows; and finally taking the logarithm of the power in each time-frequency bin. All cochleagrams were normalized using the mean and standard deviation computed from the corresponding training datasets.

**Auditory encoding models**  We fitted three different models of increasing complexity to the neural datasets: the linear-nonlinear (LN) model (the simplest model with no hidden layers) (Atencio et al., 2008), the network receptive field (NRF) model (an ANN with a single hidden layer of units) (Harper et al., 2016) and a temporal convolutional (TC) model (a convolutional neural network with a single hidden layer of units) (Pennington & David, 2023). These models have been shown to capture the response properties of A1 neurons, with those incorporating increasingly complex non-linear mechanisms demonstrating superior performance in capturing responses to natural sounds. At each time-step $t$, the models take a cochleagram-snippet $\boldsymbol{X}[t] \in \mathbb{R}^{C \times \tilde{T}}$ as input and output a response prediction $o_i^{\text{model}}[t] \in \mathbb{R}$ of the $i^{th}$ neuron as:

$$\textbf{LN} \quad o_i^{\text{LN}}[t] = f(h_i^{\text{LN}}[t]), \quad h_i^{\text{LN}}[t] = \sum_{cl} W_{icl}^{\text{LN}} X_{cl}[t] \tag{1}$$

$$\textbf{NRF} \quad o_i^{\text{NRF}}[t] = f(\sum_{j} W_{ij}^{\text{NRF}\,(2)} h_j^{\text{NRF}}[t]), \quad h_j^{\text{NRF}}[t] = f(\sum_{cl} W_{jcl}^{\text{NRF}\,(1)} X_{cl}[t]) \tag{2}$$

$$\textbf{TC} \quad o_i^{\text{TC}}[t] = f(\sum_{jl'} W_{ijl'}^{\text{TC}\,(2)} h_{jl'}^{\text{TC}}[t]), \quad h_{jl'}^{\text{TC}}[t] = f(\boldsymbol{W}_j^{\text{TC}\,(1)} * \boldsymbol{X}[t]) \tag{3}$$

The weights have the shapes $\boldsymbol{W}^{\text{LN}} \in \mathbb{R}^{N \times C \times L}$; $\boldsymbol{W}^{\text{NRF}\,(1)} \in \mathbb{R}^{H \times C \times L}$, $\boldsymbol{W}^{\text{NRF}\,(2)} \in \mathbb{R}^{N \times H}$; and $\boldsymbol{W}^{\text{TC}\,(1)} \in \mathbb{R}^{H \times C \times L}$, $\boldsymbol{W}^{\text{TC}\,(2)} \in \mathbb{R}^{N \times H \times \tilde{L}}$, where we set the input weight span of all models to $L = 50$ (200ms), the number of hidden units in the NRF and TC models to $H = 200$, and the TC model's hidden-layer's weight span to $\tilde{L} = 5$ (20ms). [1]  All layers also include a learnable bias (omitted for notational brevity) and we used the softplus function for $f$. We extended all of the encoding models to also include a gating mechanism at the outputs to modulate the response

---

[1]For the LN and NRF model $\tilde{T} = L$ and for the TC model $\tilde{T} = L + \tilde{L} - 1$.

predictions as:

$$\textbf{Gated-LN} \quad o_i^{\text{Gated-LN}}[t] = o_i^{\text{LN}}[t]\sigma(\sum_{cl} W_{icl}^{\text{LN-Gating}} X_{cl}[t]) \quad (4)$$

$$\textbf{Gated-NRF} \quad o_i^{\text{Gated-NRF}}[t] = o_i^{\text{NRF}}[t]\sigma(\sum_{j} W_{ij}^{\text{NRF-Gating}} h_j^{\text{NRF}}[t]) \quad (5)$$

$$\textbf{Gated-TC} \quad o_i^{\text{Gated-TC}}[t] = o_i^{\text{TC}}[t]\sigma(\sum_{jl'} W_{ijl'}^{\text{TC-Gating}} h_{jl'}^{\text{TC}}[t]) \quad (6)$$

where $\boldsymbol{W}^{\text{LN-Gating}} \in \mathbb{R}^{N \times C \times L}$, $\boldsymbol{W}^{\text{NRF-Gating}} \in \mathbb{R}^{N \times H}$ and $\boldsymbol{W}^{\text{TC-Gating}} \in \mathbb{R}^{N \times H \times \tilde{L}}$ denote the gating weights and $\sigma$ is the sigmoid function.

**Model training and validation** We trained each model to predict all neural responses by minimizing the negative Poisson log-likelihood between the predicted $\boldsymbol{o}^{\text{model}} \in \mathbb{R}^{N \times T}$ and target $\boldsymbol{r} \in \mathbb{R}^{N \times T}$ responses

$$\mathcal{L}_{\text{model}} = -\sum_{t,i} \left( o_i^{\text{model}}[t] \ln(r_i[t]) + r_i[t] \right) + \lambda_{\text{model}} \sum_{\boldsymbol{W} \in \mathbb{W}^{\text{model}}} \|\boldsymbol{W}\|_1 \quad (7)$$

To avoid overfitting, we also included an L1 penalty on all of the model weights $\mathbb{W}^{\text{model}}$ (besides the bias terms), weighted by hyperparameter $\lambda_{\text{model}}$. The optimal $\lambda_{\text{model}}$ was determined using five-fold cross-validation within the training dataset over a range of log-spaced values and each model was finally trained with its optimal $\lambda_{\text{model}}$ on the entire training dataset. This was repeated five times using a different random weight initialization, with the model's performance reported across all of its final fits. Training was conducted using the Adam optimizer (Kingma & Ba, 2014) with a learning rate of 0.001 over 1200 epochs and a batch size of up to 64 sound clips (64 sound clips for the awake dataset and full batch-mode for the anesthetized dataset). Lastly, we evaluated model performance using the normalized correlation coefficient $\text{CC}_{\text{norm}}$, which assesses performance independently to neural noise (Schoppe et al., 2016; Hsu et al., 2004). This measure calculates the Pearson correlation coefficient between the predicted and recorded neural responses, normalized by the maximum obtainable correlation coefficient of each neuron. We measured the peak response performance by calculating the mean squared error (MSE) between the predicted and target responses during the periods where each neurons firing rate was three standard deviations ($3\sigma$-threshold) above its mean firing rate (Harper et al., 2016). All encoding models and training were implemented using PyTorch (Paszke et al., 2019) and DevTorch (Taylor, 2024).

**Feature visualization** Feature visualization is a technique for constructing the inputs that maximally activate specific units within a neural network, by iteratively learning the inputs via gradient descent, while keeping the network weights fixed (Erhan et al., 2009; Olah et al., 2017). In our implementation, we minimized the following objective function

$$\mathcal{L}_{\text{FV}} = -\sum_{i} o_i^{\text{Gated-TC}}(\tanh(\tilde{\boldsymbol{X}}_i)) + \lambda_{\text{FV}}\|\tilde{\boldsymbol{X}}\|_2 \quad (8)$$

where we passed the trainable input weights $\tilde{\boldsymbol{X}} \in \mathbb{R}^{N \times C \times L}$ through the tanh function (to bound the values) and applied an L2 regularization constraint on the input weights (to promote smoothness). We used the Adam optimizer for training with a learning rate of $10^{-1}$ over 500 epochs.

**Dynamic STRFs** We calculated the Jacobian matrix $\partial o_i^{\text{Gated-TC}}/\partial X_{cl}$ to construct the linear dependence (*i.e.* STRF) of the neural network's predicted response of the $i^{th}$ neuron to the spectrogram input, where the sequence of such matrices is referred to as the dynamic-STRF (DSTRF) (Keshishian et al., 2020). All of the DSTRFs were calculated from the held-out test datasets using PyTorch's automatic differentiation functionality. We constructed 2D UMAP embeddings (McInnes et al., 2018) of the DSTRFs to qualitatively inspect how their spectrotemporal tuning cluster in time.[2] We also quantified the variability in tuning, by applying PCA to each neuron's DSTRF, and calculating its tuning variation as $\sum_i^{100} iv_i$, where $v_i$ is the ratio of variance explained by the $i^{th}$ principal component (PC). A low value implies less change in tuning (*i.e.* most variance is captured

---

[2]Using hyperparameters `n_neighbors` $= 100$, `min_dist` $= 0.2$ and `spread` $= 0.8$.

by the first PCs), whereas a larger value implies more change in tuning in time. Lastly, we quantified the fraction of time a neuron's DSTRF was active by counting the number of time steps where its magnitude exceeded one standard deviation. The magnitude at each time step was defined as the standard deviation of the DSTRF at that time point (Keshishian et al., 2020).

**Analyzing population-coding using SAEs** We analyzed the dimensionality of the awake and anesthetized population responses from the held-out test datasets in two ways: 1. by applying PCA to the population responses to inspect the variance explained by each principal component; and 2. by fitting an overcomplete sparse autoencoder (SAE) (Olshausen & Field, 1997) to reconstruct the population responses to examine the trade-off between reconstruction accuracy (using the $\text{CC}_{\text{norm}}$) and the required number of decoder vectors (measured as the average L0-norm of the SAE's hidden activity). The SAE was constructed as:

$$\tilde{r}_i[t] = W_{ik}^{\text{SAE}(2)} h_k^{\text{SAE}}[t], \quad h_k^{\text{SAE}}[t] = \text{ReLu}(W_{kj}^{\text{SAE}(1)} r_j[t]) \tag{9}$$

with encoding $\boldsymbol{W}^{\text{SAE}(1)} \in \mathbb{R}^{H_{\text{SAE}} \times N}$ and decoding weights $\boldsymbol{W}^{\text{SAE}(2)} \in \mathbb{R}^{N \times H_{\text{SAE}}}$, where we set the number of SAE hidden units to $H_{\text{SAE}} = 10N$. The SAE was trained to minimize a reconstruction and sparsity term, weighted by $\lambda_{\text{SAE}}$:

$$\mathcal{L}_{\text{SAE}} = \sum_{i,t} (\tilde{r}_i[t] - r_i[t])^2 + \lambda_{\text{SAE}} \|\boldsymbol{h}^{\text{SAE}}[t]\|_1 \tag{10}$$

We fitted the SAE across a range of log-spaced values for $\lambda_{\text{SAE}}$ to construct the accuracy-sparsity trade-off plots. Training was conducted using the Adam optimizer over 1000 epochs, using a batch size of 64 and a learning rate of 0.001. All responses were normalized, and we fitted the SAE on multiple random subsamples of neurons from the awake dataset (73/185) to match the number of neurons from the anesthetized dataset.

## 3 RESULTS

### 3.1 IMPROVING RESPONSE PREDICTION IN AWAKE AND ANESTHETIZED AUDITORY CORTEX VIA GATING

We fitted prior auditory encoding models of increasing complexity to the awake and anesthetized auditory recordings, including the LN (Atencio et al., 2008), NRF (Harper et al., 2016) and TC (Pennington & David, 2023) models. Additionally, we extended each of these models with a gating mechanism that modulates the predicted responses by multiplicatively adjusting the output. This extension was motivated by two factors: 1. in neuroscience, auditory cortex responses are known to be temporally modulated (*i.e.* changing in response amplitude to the same input stimulus depending on recent stimulus history) (Willmore & King, 2023), and 2. in machine learning, gating mechanisms have been shown to improve neural network performance on sequential tasks (Cho, 2014; Dauphin et al., 2017; Shazeer, 2020).

We qualitatively observed that the TC model with the gating mechanism provided the best match to both awake and anesthetized neural responses, particularly in capturing peak firing rates compared to the other models (Fig. 1a). To quantify prediction performance, we used the normalized correlation coefficient (CCnorm), where CCnorm = 0 represents chance-level prediction and $\text{CC}_{\text{norm}} = 1$ indicates a perfect fit. Including the gating mechanism improved the performance of all models, with the gated-TC model achieving the highest performance on both the awake (CCnorm = 0.64) and anesthetized (CCnorm = 0.75) datasets (Fig. 1b). We found the inclusion of the gating mechanism in the TC model to reduce the peak activity MSE over all neurons on both the awake (3.6%) and anesthetized (4.4%) datasets (Fig. 1c). Thus, the reduction in prediction error during the peak activity periods appears to be the drive in performance of the gating mechanism. However, as this improvement was largely similar between the two datasets, suggests that anaesthesia does not disrupt the biological gating mechanisms within cortical auditory neurons.

All models predicted responses in the anesthetized state more accurately than in the awake state (Fig. 1b). The gated-TC model showed a larger relative performance increase over the gated-LN model in the awake condition (31%) compared to the anesthetized condition (13%). As the gated-TC model contains more non-linear operations (*i.e.*, convolutions and stacking of layers) compared

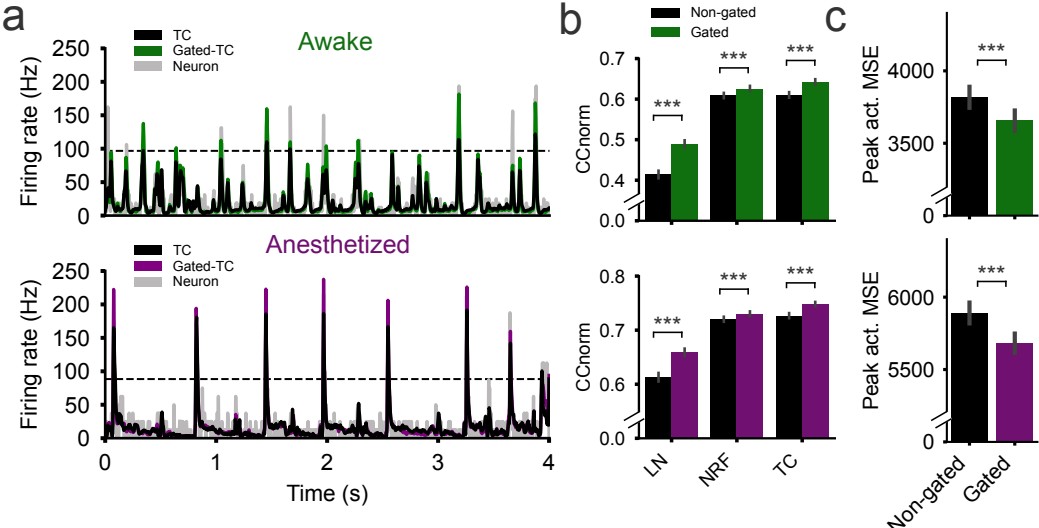

Figure 1: **Gating improves the response prediction of awake and anesthetized auditory cortical neurons across different encoding models. a.** Neural response prediction of the temporal-convolution (TC) (black) and gated temporal-convolution model (green / purple) for one example neuron (gray) from the awake (top) and anesthetized (bottom) datasets. The horizontal dotted black line indicates the $3\sigma$-threshold used to identify large peaks in the response. **b.** Comparison of the prediction quality of different encoding models and their gated extensions on the awake (top) and anesthetized (bottom) datasets. **c.** Average peak activity MSE of the non-gated- and gated-TC model on the awake (top) and anesthetized (bottom) datasets. Bars plot the mean and standard error over the model units, with statistical significance assessed using the Wilcoxon signed-rank test (***p<0.001).

to the gated-LN model, suggests that the conscious auditory cortex engages more complex non-linear processes than the unconscious auditory cortex. Given its superior performance, we selected the gated-TC model to further investigate the differences between awake and anesthetized auditory responses in subsequent analyses.

## 3.2 VISUALIZING NON-LINEAR SPECTROTEMPORAL TUNING OF AWAKE AUDITORY CORTICAL NEURONS

Spectrotemporal tuning of auditory neurons is typically assessed using the spectrotemporal receptive field (STRF) (the weights of the LN model) (Atencio et al., 2008; Rabinowitz et al., 2011). In our analysis, the STRFs of the anesthetized neurons exhibited distinct tuning patterns, such as excitation near the present followed by lagging inhibition into the past (DeCharms et al., 1998). In contrast, many of the STRFs of the awake neurons generally lacked clear tuning. To further investigate the tuning properties of the awake neurons, we used feature visualization (FV) (Olah et al., 2017). This is an interpretability technique often used to study the tuning properties of units in convolutional neural networks trained on images, where neural network inputs are iteratively optimized via gradient descent to maximally activate particular unit responses. After optimization, these learnt input weights give an estimate of a units tuning preference. As commonly done (Olah et al., 2017), we applied an L2 smoothing regularization constraint to the input weights during training, where different levels of regularization can drastically influence the structure of the learnt input weights (Appendix Fig. 3).

We observed that the FV-STRFs of the anesthetized neurons were more strongly correlated with their corresponding LN-STRFs (CC = 0.34), compared to the awake neurons (CC = 0.21), across a range of smoothing regularizations (Fig. 2a). The FV-STRFs of the anesthetized neurons closely resembled the LN-STRFs. In contrast, we found that the LN-STRFs of the awake neurons that did not exhibit clear tuning properties reveal a more distinctive spectrotemporal structure in their FV-

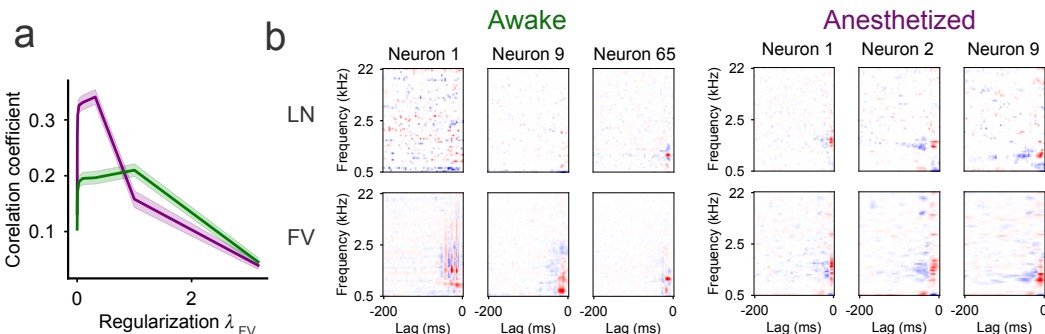

Figure 2: **Visualizing spectrotemporal tuning of non-linear A1 neurons using feature visualization. a.** Correlation coefficient between the STRFs obtained from the LN-model and feature visualization (FV) method across different L2 regularization $\lambda_{\text{FV}}$. Lines and shaded region plot the mean and standard error over neurons from the awake (green) and anesthetized (purple) datasets. **b.** STRFs obtained from the LN-model (top) and FV method (bottom) for different example neurons from the awake (left) and anaesthetized (right) datasets. Red corresponds to excitation and blue to inhibition.

STRFs, such as broadband tuning with lagging inhibition or striped zebra-like patterns of excitation and inhibition (Fig. 2b). These findings suggest that awake A1 neurons - similar to anesthetized neurons - are maximally driven by specific spectrotemporal patterns. However, the notable differences between the FV-STRFs and LN-STRFs in the awake condition indicate that the tuning of awake A1 neurons may be more dynamic and likely changing over time (as their tuning is not readily captured by a fixed spatiotemporal pattern).

### 3.3 AWAKE A1 NEURONS EXHIBIT DYNAMIC SPECTROTEMPORAL TUNING WHICH ANESTHESIA SWITCHES OFF

FV-STRFs provide a clear interpretation of the stimuli that maximally drive an auditory neuron's response at a fixed point in time, but they do not capture how tuning evolves over time. Dynamic-STRFs - a neural network linearization technique - have been used to study the dynamic tuning properties of primary and non-primary auditory cortical neurons from electrocorticography recordings in neurosurgical patients with epilepsy (Keshishian et al., 2020). We used this technique to examine the differences in dynamic tuning between electrophysiologically recorded responses in awake and anesthetized A1 neurons, using the predicted responses of the gated-TC model.

We found the DSTRFs of the awake neurons to exhibit dynamic spectrotemporal tuning in time (Fig. 3a). When visualized in a two-dimensional space using UMAP, the dimensionality-reduced representations of the DSTRFs formed distinct clusters, each corresponding to different spectrotemporal tunings. Applying PCA to each DSTRF showcased the variance in each DSTRF to be dispersed across multiple principal components (PCs). In contrast, the DSTRFs of the anesthetized neurons showed a fixed spectrotemporal structure over time. The UMAP visualization did not reveal distinct clusters, and the majority of the DSTRF variance was concentrated in the first PC dimension (Fig. 3b).

To quantify these observations across the neuron population, we measured each neuron's DSTRF tuning variability (the variability in spectrotemporal tuning) and activity (the fraction of time the dynamic-STRF is "on") (see Methods). We found that awake neurons exhibited significantly greater tuning variability (Fig. 3c) and activity (Fig. 3d) compared to the anesthetized neurons. To address the potential confound of the awake dataset containing more training stimuli than the anesthetized dataset, we retrained the gated-TC model on a uniformly sampled subset of awake stimuli, matched in number to the anesthetized training stimuli. We found the DSTRFs to still exhibit dynamic tuning (Appendix Fig. 4) and the differences in tuning variability and activity to persist (Appendix Fig. 5), although the tuning variability was slightly reduced. Further subsampling the number of stimuli and neurons to be matching also did not change the significance in the tuning variability and activity (Appendix Fig. 6). Lastly, comparing the tuning measures on the same datasets sounds with an

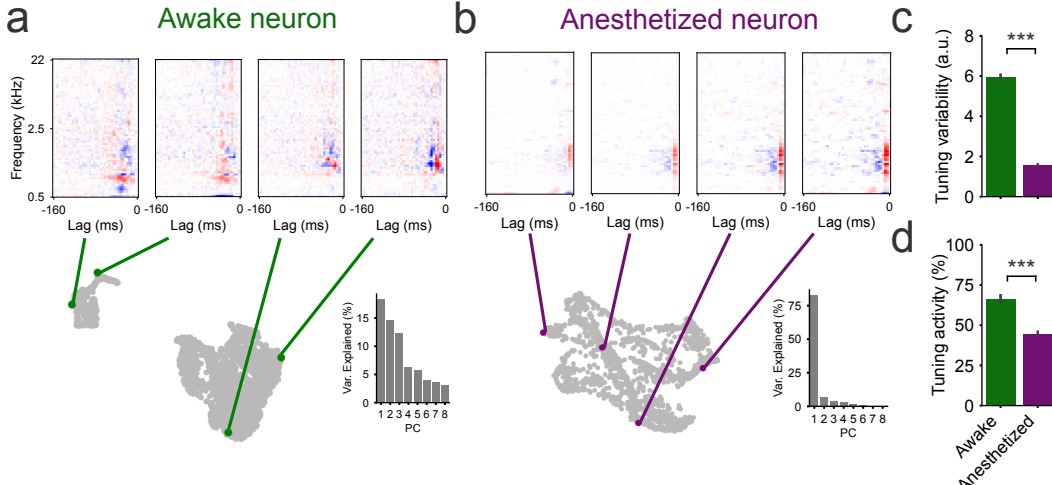

Figure 3: **Awake A1 neurons exhibit dynamic spectrotemporal tuning which anesthesia switches off. a.** Dynamic-STRF of an example awake A1 neuron showcasing changing spectrotemporal tuning in time (top) with the UMAP-projection of all of the neuron's STRFs (bottom left) and their scree-plot (bottom right). Red corresponds to excitation and blue to inhibition. **b.** Dynamic-STRF of an example anesthetized A1 neuron showcasing a fixed spectrotemporal tuning in time. As the spectrotemporal tuning is stationary in time, the UMAP does not exhibit distinct clusters and most of the variance of the dynamic-STRF is captured by its first principal component. **c.** Tuning variation (the variability in spectrotemporal tuning) and **d.** tuning activity (the fraction of time the dynamic-STRF is "on") of the awake and anesthetized A1 neurons. Bars plot the mean and standard error over neurons, with statistical significance assessed using the Mann-Whitney U test (***p<0.001).

unmatched and matched number of training samples, demonstrates the awake neurons to still exhibit a significantly higher tuning variability (Appendix Fig. 7). Although interestingly, the activity between the awake and anesthetized A1 neurons became more similar. These findings suggest that anesthesia shifts A1 neurons from polysemantic (responding to different spectrotemporal inputs) to monosemantic (responding to a fixed spectrotemporal input) tuning, however it is less clear if anesthesia affects the fraction of time that the dynamic-STRF is "on".

### 3.4 THE CONSCIOUS AUDITORY CORTEX EXHIBITS A HIGHER-DIMENSIONAL POPULATION CODE THAN THE UNCONSCIOUS AUDITORY CORTEX

Having identified different coding properties at the individual neuron level, we sought to understand how these characteristics influence the population code - the collective response of neurons in primary auditory cortex. We examined this question by analyzing the dimensionality of the population code, where lower dimensionality indicates more redundancy and less capacity for encoding, while higher dimensionality suggests greater diversity and complexity for neuronal encoding of diverse sounds.

We first applied PCA to the trial-averaged population responses from the held-out test datasets (Fig.4a) and found the awake population code to require more principal components to capture the same amount of variance compared to the anesthetized neural population (Fig.4b). We made a similar observation when applying PCA to the trial-averaged population responses from the training datasets (Appendix Fig. 8). This suggests that the awake neural population exhibits a higher dimensionality. To corroborate this observation, we examined the population codes using sparse autoencoders (SAEs) (Fig.4c), which serve as a normative model of brain function (Olshausen & Field, 1997) and are increasingly being used to interpret the hidden-activity of large language models (Cunningham et al., 2023).

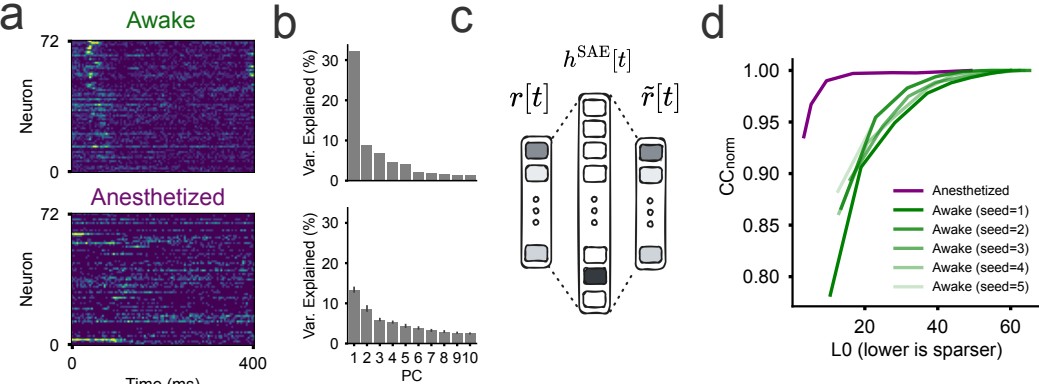

Figure 4: **Dimensionality of the neural population code of awake and anesthetized A1 neurons. a.** Example snippets of the neural population code to natural stimulus sounds in awake (top) and anesthetized (bottom) ferrets (light colour corresponds to activity). **b.** Corresponding percentage of variance explained per PC-dimension of the awake and anesthetized neural population codes. **c.** Schematic of the sparse autoencoder outputting prediction $\tilde{r}[t]$ of the neural population activity $r[t]$ at time $t$, whose sparsity in hidden-activity $h^{\text{SAE}}[t]$ gives a measure of population-dimensionality. **d.** Trade-off between the SAE's population-code reconstruction (y-axis) over different levels of activity sparsity (x-axis) for the awake (green) and anesthetized (purple) population codes. We matched the number of neurons in the awake ($n = 183$) and anaesthetized ($n = 73$) datasets by randomly subsampling five different subsets of $n = 73$ awake neurons for the analyses is **b.** and **d.**. Bars in **b.** plot the mean and standard error over neurons.

SAEs are autoencoders with an overcomplete basis (*i.e.* more hidden units than input/output units) trained to reconstruct input data under sparsity constraints on the hidden-activity. We trained SAEs on the neural population responses and analyzed the trade-off between reconstruction accuracy and the average number of active hidden units. We took the average number of active hidden units in the SAE to be a proxy of the dimensionality of the population code, as each SAE hidden-state (*i.e.* decoder vector) is thought to represent a different variable/feature within the networks hidden activity (Cunningham et al., 2023). Although technically, we slightly abuse the conventional definition of the dimensionality of a space here, as the hidden states of the SAE may be non-orthogonal. We found the awake neural population to require a larger number of decoder vectors to achieve the same reconstruction accuracy as the anesthetized population, demonstrating the awake neurons to exhibit a higher-capacity coding regime than the anesthetized neurons (Fig.4d). To facilitate a more direct comparison between the awake and anesthetized datasets, we randomly subsampled neurons from the awake dataset to match the number of neurons in the anesthetized dataset.

### 3.5 ANESTHESIA DECOUPLES CORTICAL CONNECTIVITY

The question remains how anesthesia transforms the high-dimensional population code of polysemantic neurons in the auditory cortex into a lower-dimensional code of monosemantic neurons. Anesthesia is known to suppress cortical connectivity (Voss et al., 2019). To investigate if disruptions in connectivity underpin our observed differences in sensory processing and population coding, we analyzed the number of incoming connections to the output units in the gated-TC model trained on the awake and anesthetized datasets.

We quantified the incoming connections to each output unit by counting the number of absolute weight values exceeding a threshold of $\sigma$ standard deviations above their mean. We observed the model trained on the awake dataset to have more incoming connections per output unit compared to the model trained on the anesthetized dataset (Fig. 5a), finding a statistically significant difference in the number of connections across various threshold values (Fig. 5b). In contrast, the magnitude of the weights was largely similar between the models across the various threshold values, apart from the highest threshold values (Fig. 5c). These findings suggest that our observed differences in sensory processing and population coding between the conscious and unconscious auditory cortex

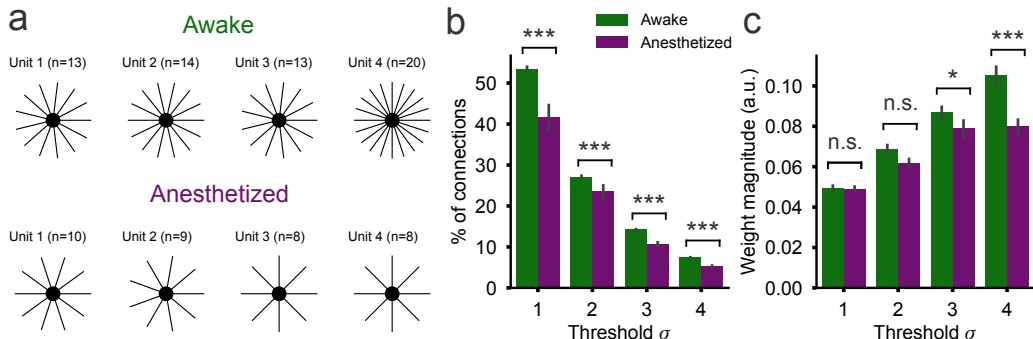

Figure 5: **Quantifying the number of connections in the gated temporal-convolution model trained on the awake and anesthetized datasets. a.** The number of incoming connections to different example output units from the gated-TC model trained on the awake (top) and anesthetized (bottom) datasets (using a threshold of $\sigma = 4$). **b.** Number of incoming connections to the output units and **c.** their magnitude across different threshold values $\sigma$. Bars plot the mean and standard error over the model units, with statistical significance assessed using the Mann-Whitney U test (*p<0.05, ***p<0.001).

are the result of anesthesia decoupling cortical connectivity, as opposed to shifting connectivity strengths.

## 4 DISCUSSION

We explored how general anesthesia affects sensory processing in the primary auditory cortex (A1) using auditory encoding models trained to predict neuronal responses to natural sounds in awake and anesthetized ferrets. Utilizing mechanistic interpretability techniques, we discovered that anesthesia induces a shift in neuronal tuning from polysemantic - where neurons respond to multiple spectrotemporal features - to monosemantic, responding to a single spectrotemporal input. Furthermore, we found this transition to reduce the dimensionality of the population code due to a decoupling of connectivity in the models.

**Gated encoding models of auditory cortex** The gating mechanism in neural networks modulates the activations of units and has been shown to enhance the capacity of recurrent (Hochreiter, 1997; Cho, 2014), convolutional (Dauphin et al., 2017) and Transformer network architectures (Shazeer, 2020; Hua et al., 2022). Similarly, we found that extending standard encoding models of the auditory cortex with a gated-output to improve response predictions to natural sounds, by multiplicatively scaling the model output. This mechanism likely captures the time-varying sensitivity of sensory neurons to input stimuli (Weber et al., 2019; Willmore & King, 2023), where neurons within the auditory pathway have been shown to adjust their responses to changes in mean sound level (Dean et al., 2005), sound contrast (Rabinowitz et al., 2011); and more complex features like timbre (Piazza et al., 2018) and spatial-statistics (Gleiss et al., 2019). Such adaptive mechanisms are believed to expand the sensory encoding range to accommodate a broader spectrum of stimulus values (Barlow et al., 1961; Laughlin, 1981). We found the gating mechanism to reduce response prediction error during periods of peak activity, which likely explains its performance boost. Thus, the gating mechanism likely "amplifies" the sensitivity of the auditory encoding models to particular stimuli - in contrast to traditional neural network models like the LSTM in which the gating mechanism rather "forgets" particular sensory input. Although the gating mechanism improves response prediction, it is still far from perfectly capturing the response properties of cortical A1 neurons to natural sounds. The missing gap in performance may be attributed to missing non-linear mechanisms in the model, not enough training data or due to internal biological states which cannot be recorded. Lastly, it would be interesting to explore other output mechanisms in future work such as dynamic thresholding, of only outputting a predicted response when a certain criterion is met. This may boost performance by filtering out less relevant signals that the gating mechanism may not capture.

**Anesthesia induced notable changes in auditory cortical processing**   How anesthesia affects sensory processing is still not completely understood. We found that in awake ferrets, neurons in the primary auditory cortex exhibit polysemantic spectrotemporal tuning, responding to multiple spectrotemporal features. Under anesthesia, neurons shift to monosemantic tuning, preferring a fixed spectrotemporal input. This shift aligns with previous observations: distinguishing complex sound classes becomes difficult under anesthesia (Plourde et al., 2006), auditory processing in associated cortical areas is disrupted (Krom et al., 2020), and A1 neurons become less selective to specific frequency ranges (Gaese & Ostwald, 2001). Our findings reconcile these observations, as reduced dynamic tuning under anesthesia could explain decreased selectivity and processing disruptions. We also observed that the population code in A1 exhibits reduced dimensionality under anesthesia, indicating decreased cortical encoding capacity. This corroborates Noda & Takahashi (2015), who reported increased noise correlations in the auditory cortex under anesthesia, leading to more redundant neural activity.

**Disruptions to cortical connectivity: top-down or bottom-up?**   We found that our neural network model trained on the awake dataset had more incoming connections per output unit compared to the model trained on the anesthetized dataset. This suggests that the changes in spectrotemporal tuning and population coding of A1 neurons under anesthesia may result from a decoupling of cortical connectivity. However, it remains unclear whether these decoupled connections correspond to disruptions in corticocortical (top-down) or thalamocortical (bottom-up) pathways, as both have been shown to be suppressed during general anesthesia (Voss et al., 2019). Given that our model uses a strictly feedforward architecture, it primarily captures bottom-up processes. Therefore, our results support the notion that the changes in A1 neuron tuning and population coding during anesthesia result, at least in part, from disruptions in thalamocortical connectivity. This interpretation is consistent with the findings of Schumacher et al. (2011), who reported that the excitability of auditory midbrain neurons is reduced during anesthesia, which would result in less afferent input to the cortical neurons. Nevertheless, it is also possible that disruptions in top-down cortical inputs also contribute to the observed changes under anesthesia. Future studies incorporating recurrent architectures could help elucidate the relative contributions of top-down and bottom-up pathways to the alterations in auditory processing during anesthesia.

**Experimental limitations**   Although we used neural recordings from the same sensory region (A1) and species (ferret), the awake and anesthetized recordings were obtained from different labs using different animals and in response to different sets of natural sounds. This introduces potential confounds related to individual variability and stimulus differences that could affect the results. However, both datasets include similar natural sounds of human speech, animal vocalizations, and environmental sounds, which we found to exhibit similar average power spectra between the datasets (Appendix Fig. 1). Additionally, both datasets exhibit similar tuning properties, where we found the averaged population STRF of the neurons in the awake and anesthetized datasets to exhibit a temporally asymmetric power profile with excitation near the present followed by lagging inhibition into the past (Appendix Fig. 9), as has similarly been reported in prior work (DeCharms et al., 1998).

Although the datasets were recorded using different hardware and spike-sorting software, they have previously been used in prior work for side-by-side comparison, where Rahman et al. (2020) used these datasets to contrast the performance of various cochleagram models in predicting auditory cortical responses under awake and anesthetized conditions. Although the anesthetized recordings were done using silicon probe electrodes and the awake recordings were done using tungsten microelectrodes, these different systems have previously been shown to provide comparable action potential recordings (Saha et al., 2010). Furthermore, both datasets were gathered from reputable labs, which have likely calibrated their recording systems to be as precise as possible. Thus, it seems unlikely that the differences between the different labs' experimental setups would explain the auditory tuning differences we report. Where possible, we attempted to mitigate the potential confounding issues in the dynamic spectrotemporal tuning analysis (section 3.3) by qualitatively comparing DSTRFs (Appendix Fig. 4), by matching the number of training stimuli between the datasets (Appendix Fig. 5); by matching the number of training stimuli and neurons between the datasets (Appendix Fig. 6); and by comparing the tuning measures on the same datasets with an unmatched and matched number of training samples (Appendix Fig. 7). We also matched the number of neurons in the population-coding dimensionality analysis (section 3.4). Future follow-up work should ideally obtain recordings in the same animals to the same sounds.

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

## A APPENDIX

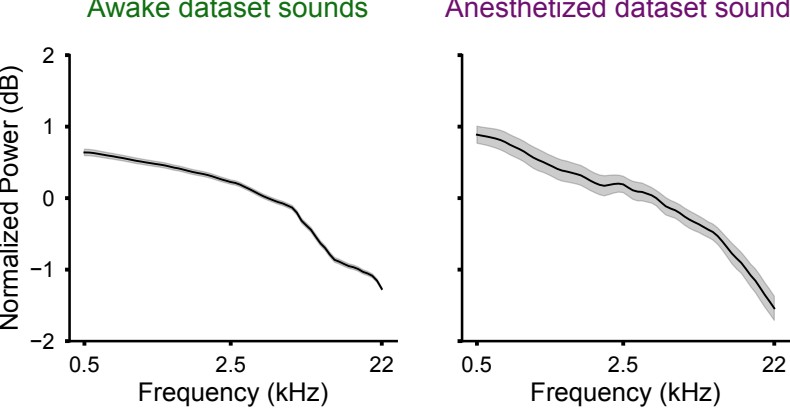

Figure 1: **Power spectrum of the natural sounds from the awake and anesthetized datasets.** Solid line plots the mean and the shaded area corresponds to the standard error of the power over time and sound clips.

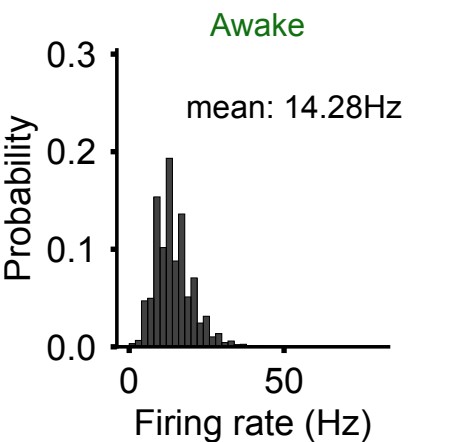
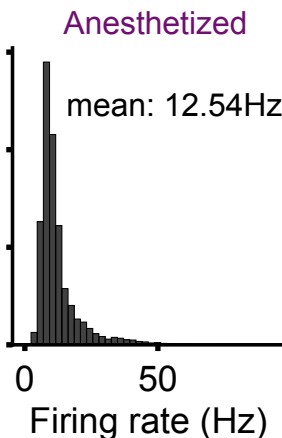

Figure 2: **Firing rate distribution.** Firing rate distribution of the A1 neurons in the awake (left) and anesthetized dataset (right). Here, we calculated the mean firing rate per neuron per clip and binned the resulting values.

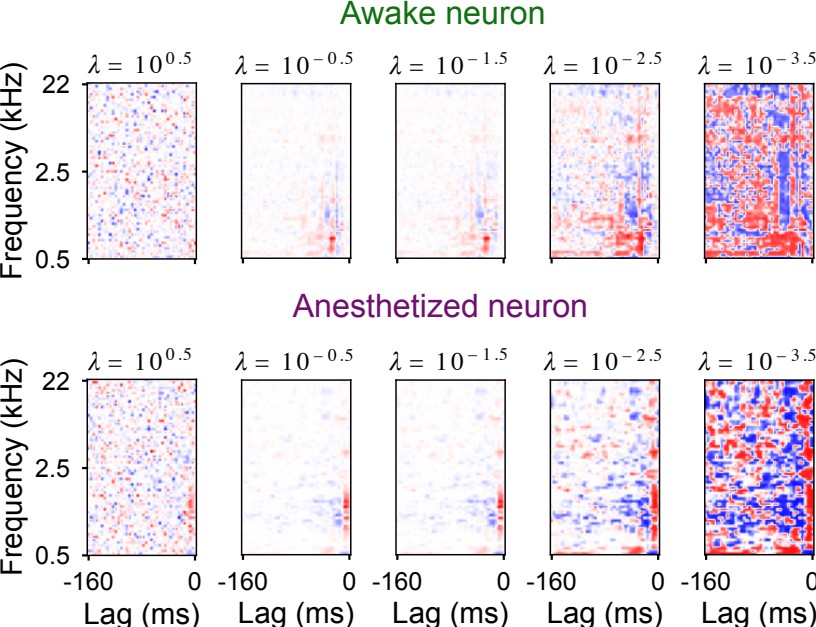

Figure 3: **Visualizing spectrotemporal tuning of non-linear A1 neurons using feature visualization across different L2 regularization values.** STRFs obtained from the FV method for an example A1 awake neuron (top) and anesthetized neuron (bottom). Notice the change in power for varying regularization values.

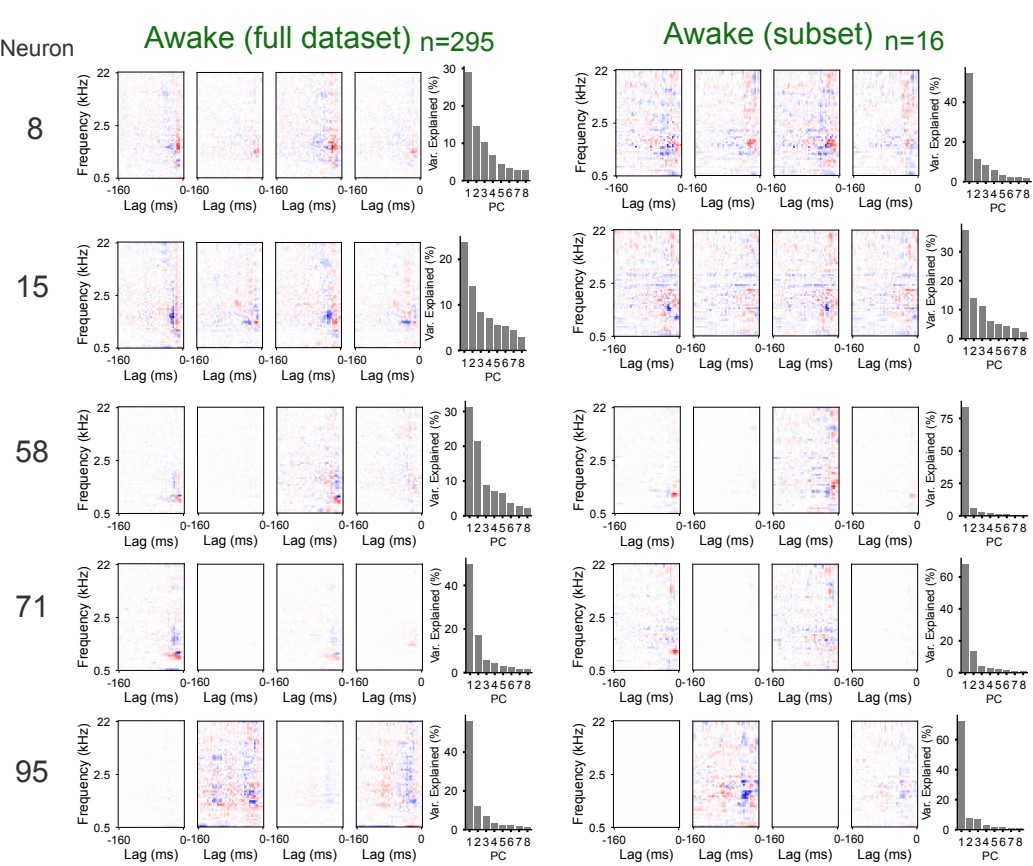

Figure 4: **Example DSTRF of awake A1 neurons when training with a different number of data samples.** Left: DSTRFs obtained from the gated-TC model trained on the full awake A1 dataset and right: DSTRFs obtained from the gated-TC model trained on a random subset of data samples matching the number of data samples of the anesthetized dataset. Scree-plot obtained from each neurons DSTRF.

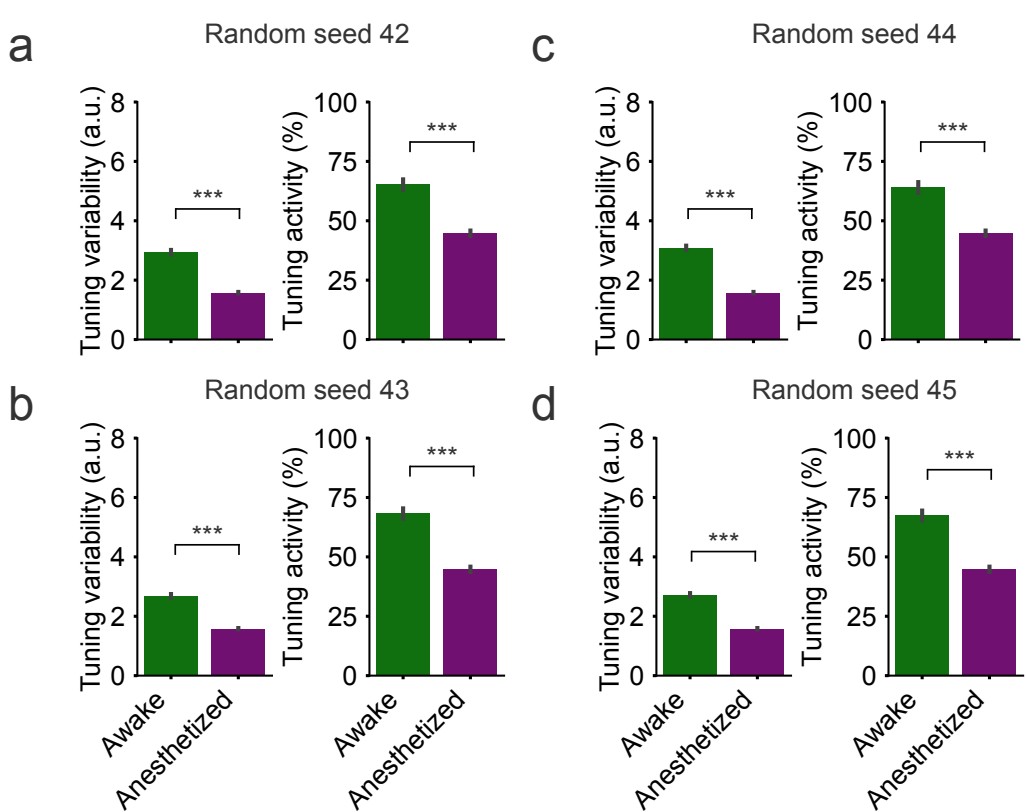

Figure 5: **Tuning variation and activity between awake and anesthetized A1 neurons for matching number of training samples between the datasets.** Tuning variation and activity calculated from the gated-TC model trained to predict the awake responses (green) and anesthetized responses (purple), but with the gated-TC model trained with fewer training stimuli for the awake responses matching the number of stimuli in the anesthetized dataset. **a.** to **d.** report the tuning metrics calculated across different random seeds used for uniformly sampling random training sounds from the awake dataset. Bars plot the mean and standard error over neurons, with statistical significance assessed using the Mann-Whitney U test (***p<0.001).

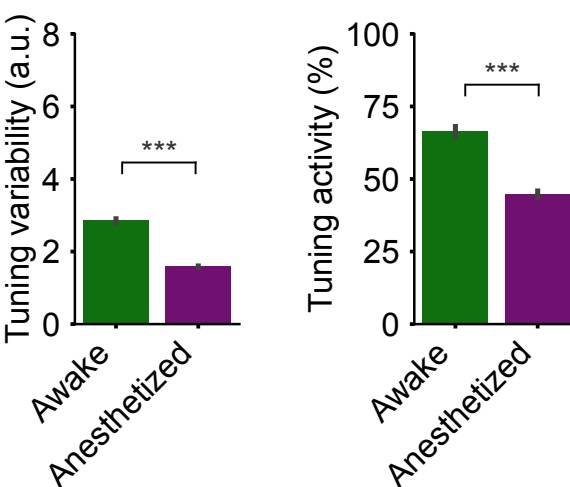

Figure 6: **Tuning variation and activity between awake and anesthetized A1 neurons for matching number of training samples and matching number of neurons between the datasets.** Tuning variation and activity calculated from the gated-TC model trained to predict the awake responses (green) and anesthetized responses (purple), but with the gated-TC model trained with fewer training stimuli for the awake responses matching the number of stimuli in the anesthetized dataset. This was repeated five times, each time uniformly sampling random training sounds and neurons from the awake dataset to match the number of training sounds and neurons from the anesthetized dataset. Bars plot the mean and standard error over neurons, with statistical significance assessed using the Mann-Whitney U test (***$p<0.001$).

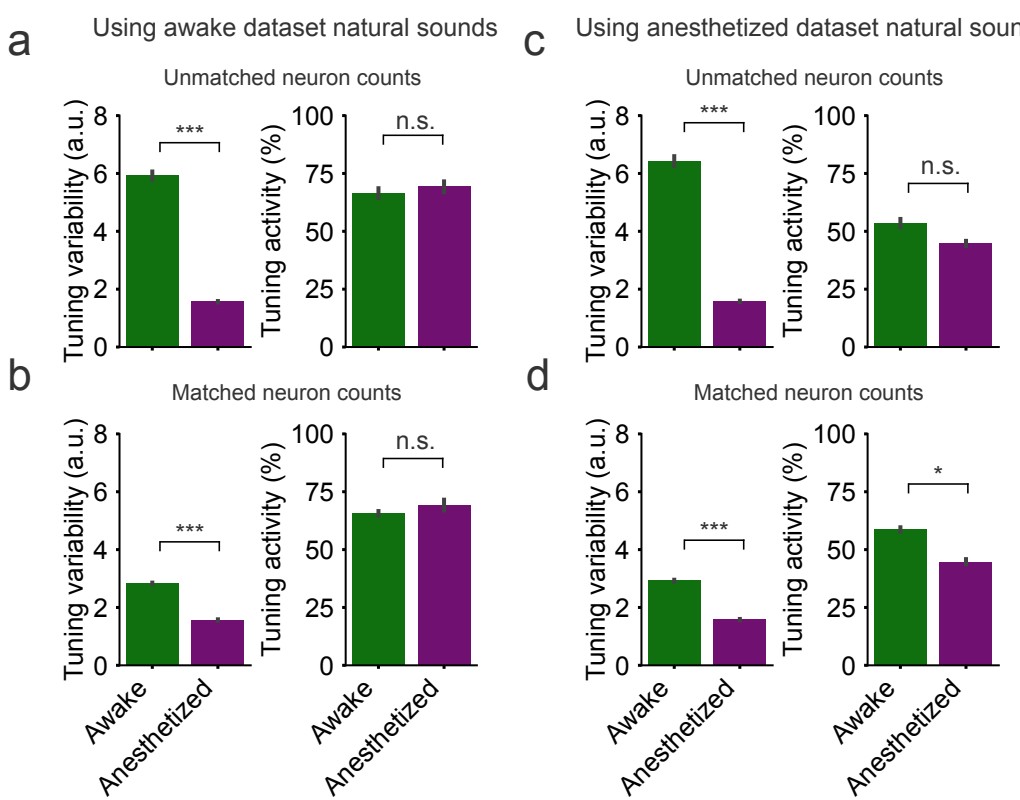

Figure 7: **Tuning variation and activity between the awake and anesthetized A1 neurons compared on the same sounds. a.** Tuning variation and activity calculated from the gated-TC model trained to predict the awake responses (green) and anesthetized responses (purple) using the awake dataset natural sounds. **b.** Same as in **a.** but with the gated-TC model trained with fewer training stimuli for the awake responses matching the number of stimuli in the anesthetized dataset (this was repeated five times, each time uniformly sampling random training sounds from the awake dataset). **c.** and **d.** are the same as **a.** and **b.** respectively, but with the tuning variation and activity calculated using the anesthetized dataset natural sounds. Bars plot the mean and standard error over neurons, with statistical significance assessed using the Mann-Whitney U test (*p<0.05, ***p<0.001).

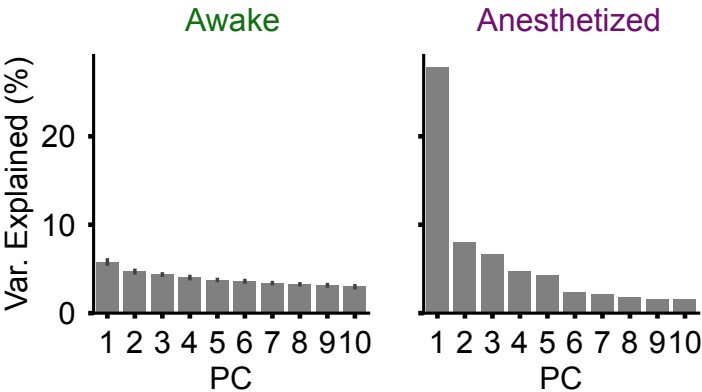

Figure 8: **Dimensionality of the neural population code of awake and anesthetized A1 neurons calculated using the training dataset responses.** Corresponding percentage of variance explained per PC-dimension of the awake and anesthetized neural population codes from the responses obtained from the training datasets. We uniformly sampled the number of neurons and the number of sounds in the awake dataset to be matched to the anesthetized dataset. This was repeated five different times. Bars plot the mean and standard error over neurons.

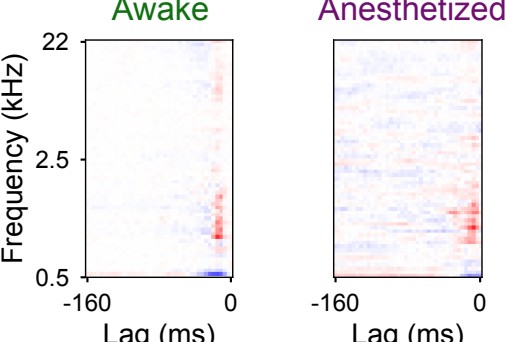

Figure 9: **Population STRFs.** Averaged STRF over the neurons in the awake (left) and anesthetized datasets (right). Notice the similar frequency tuning properties where the population averaged STRFs exhibit a temporally asymmetric power profile with excitation near the present (red) followed by slight lagging inhibition into the past (blue).

