# OpenReview forum: "Shapeshifters: Auditory cortical neurons switch from polysemantic to monosemantic under anesthesia"
_ICLR.cc/2025/Conference — Submitted to ICLR 2025_

### Official Review · Reviewer_CqcH · 2024-10-30

**Soundness:** 2
**Presentation:** 3
**Contribution:** 2
**Rating:** 5
**Confidence:** 3

**Summary:**

Authors analysed and modelled the responses of neurons from the primary auditory cortex (A1) in awake and anesthetised ferrets. They compared the performance of several recent decoding models and found the temporal convolutional model to be the best among tested alternatives. Further, authors showed a significant improvement in performance of encoding models by adding gating of the output. They found that neurons in awake animals are characterised by stronger, more dynamic and more complex tuning of single neurons compared to anesthetised animals. Also, they found that neural populations in awake animals have higher dimensionality compared to anesthetised animals. Finally, they find that in the awake animals, neurons receive more synaptic connections and are therefore more strongly coupled than in anesthetised animals.

**Strengths:**

The paper combines a number of known methods and techniques to brings new insights into the encoding differences between neural activity in the auditory cortex in the awake and anesthetised brain state. The paper addresses an interesting question. The writing is transparent about the methods and about the results.

**Weaknesses:**

1) While I appreciated the effort for transparency, the clarity of writing can be improved on several places. For example, the title and content of the section 3.2 could use clearer writing. The title of section 3.2 mentions that "anestetised neurons are more interpretable that awake neurons" . This is an awkward statement, because the level of interpretability always depends on what is being interpreted.

2) Authors do not explain well what is the mechanism of gating and why it improves encoding. Some more methodological description of gating could be useful in Methods, and an intuitive description of what gating does would be a useful addition to Results.

3) The description of different encoding models (LN, NRF, TC) can also be improved, as it remains unclear why would we expect these models to perform differently.

4) The discussion about depression in the last paragraph of the paper reads like a long shot and is tangential to the rest of the material presented in the paper. I suggest removing it and using the space to better describe the most important results.

**Questions:**

1) Even the best model systematically underestimates (undershoots) the firing rate during the sustained response in anesthetised animals. What might be the cause?

2) The authors write that "the average number of active hidden units is a proxy of the dimensionality of the population code". This interpretation of the dimensionality of a population code is new to me, as the dimensionality is typically related to the number of independent variables encoded by a neural network. This is similar to the notion of dimensionality authors use when describing their Fig. 4b. Could authors comment on that?

---

> ### Author Response · Authors · 2024-11-26
>
> We thank the reviewer for their positive assessment, and we hope that our answers below satisfy their comments.
>
> **W1: While I appreciated the effort for transparency, the clarity of writing can be improved on several places. For example, the title and content of the section 3.2 could use clearer writing. The title of section 3.2 mentions that "anestetised neurons are more interpretable that awake neurons". This is an awkward statement, because the level of interpretability always depends on what is being interpreted.**
>
> > We agree with your assessment and have now changed the title of section 3.2 to “Visualizing non-linear spectrotemporal tuning of awake auditory cortical neurons” and have added more background and context to section 3.2. In addition, we have made an effort to increase the clarity of writing throughout our manuscript (all edits in red) and have provided several follow-up analyses in the Appendix.
>
> **W2: Authors do not explain well what is the mechanism of gating and why it improves encoding. Some more methodological description of gating could be useful in Methods, and an intuitive description of what gating does would be a useful addition to Results.**
>
> > We apologize for not making this more clear. We have now included more descriptions of the gating mechanism in the Methods, Results and Discussion. We have also run new analysis to further investigate what in particular about the gating mechanism is driving the boost in performance. We found the inclusion of the gating mechanism in the TC model to reduce the peak activity MSE over all neurons on both the awake (3.6%) and anesthetized (4.4%) datasets (see updated Fig. 1c). Thus, the reduction in prediction error during the peak activity periods appears to be the drive in performance of the gating mechanism. However, as this improvement was largely similar between the two datasets, suggests that anaesthesia does not disrupt the biological gating mechanisms within cortical auditory neurons.
>
> **W3: The description of different encoding models (LN, NRF, TC) can also be improved, as it remains unclear why would we expect these models to perform differently.**
>
> > We now provide more context in the Methods and Results section 3.1. In summary, the linear-nonlinear (LN) model is the simplest model with no hidden layers, the network receptive field (NRF) is an ANN with a single hidden layer of units, and the temporal convolutional (TC) model is a convolutional neural network with a single hidden layer of units. These models have been shown to capture the response properties of A1 neurons, with those incorporating increasingly complex non-linear mechanisms demonstrating superior performance in capturing responses to natural sounds.
>
> **W4: The discussion about depression in the last paragraph of the paper reads like a long shot and is tangential to the rest of the material presented in the paper. I suggest removing it and using the space to better describe the most important results.**
>
> > We agree with your assessment and have now removed this paragraph. We now include a summary paragraph at the beginning of our Discussion and have significantly expanded the paragraphs on the gating mechanism and experimental limitations.
>
> **Q1: Even the best model systematically underestimates (undershoots) the firing rate during the sustained response in anesthetised animals. What might be the cause?**
>
> > It remains an active area of research to develop models which accurately capture the responses of neurons. This underestimate may be due to missing non-linear mechanisms in the model, not enough training data or due to internal biological states which cannot be recorded. We have now expanded upon this in our Discussion.
>
> **Q2: The authors write that "the average number of active hidden units is a proxy of the dimensionality of the population code". This interpretation of the dimensionality of a population code is new to me, as the dimensionality is typically related to the number of independent variables encoded by a neural network. This is similar to the notion of dimensionality authors use when describing their Fig. 4b. Could authors comment on that?**
>
> > You are correct and we slightly abuse the definition of dimensionality here. We have attempted to make this more clear to the reader now in our Results section 3.4:  “We took the average number of active hidden units in the SAE to be a proxy of the dimensionality of the population code,  as each SAE hidden-state (i.e. decoder vector) is thought to represent a different variable/feature within the networks hidden activity (Cunningham et al. 2023). Although technically, we slightly abuse the conventional definition of the dimensionality of a space here, as the hidden states of the SAE may be non-orthogonal.”

---

> > ### Author Response · Authors · 2024-12-01
> >
> > Dear reviewer, we hope our responses have addressed your questions and clarified any concerns. As the extended discussion period comes to an end, please let us know if you have any further questions. We greatly appreciate your time and effort.

---

> > ### Comment · Reviewer_CqcH · 2024-12-02
> >
> > I thank the Authors for replying to my questions.
> >
> > Follow-up question on W2: Looking at the Fig. 1a, it seems that gating leads to higher peaks compared to the firing rate peaks of the non-gated model. This seems surprising because gating bounds the firing rate. Could authors comment on that?
> >
> > Follow-up comment on Q2: I think that the "slight abuse of conventional definition" is not necessary. Authors could simply report the result as the change in the average number of active hidden units and potentially discuss the implications of this result on dimensionality. If change in dimensionality is reported as an important result (that is even mentioned in the first paragraph of Discussion), it should be measured with appropriate technique that directly measures the dimensionality.

---

> > > ### Author Response · Authors · 2024-12-02
> > >
> > > **Follow-up question on W2: Looking at the Fig. 1a, it seems that gating leads to higher peaks compared to the firing rate peaks of the non-gated model. This seems surprising because gating bounds the firing rate. Could authors comment on that?**
> > >
> > > >This is correct, the gated model better captures the peak firing responses compared to the non-gated model. The gating mechanism multiplicatively scales the outputs by a dynamic value ranging between 0 and 1 (see Eq. 4-6). This may appear to bound the output. However, the gated and non-gated models are trained separately and hence have different weights. Thus, the outputs of the gated model are not bounded by the outputs of the non-gated model. We have now made this more explicit in the Methods with “The gated models are trained separately from their non-gated counterparts, and can thus learn to better predict neural responses by dynamically scaling output predictions.”.
> > >
> > > **Follow-up comment on Q2: I think that the "slight abuse of conventional definition" is not necessary. Authors could simply report the result as the change in the average number of active hidden units and potentially discuss the implications of this result on dimensionality. If change in dimensionality is reported as an important result (that is even mentioned in the first paragraph of Discussion), it should be measured with appropriate technique that directly measures the dimensionality.**
> > >
> > > > We agree and have removed this statement from section 3.4. PCA is used as a more appropriate measure of dimensionality (Fig. 4b).
> > >
> > > Thank you for your valuable feedback in improving our submission. Please feel free to let us know if you have any further questions or concerns, and we will be happy to address them.

---

> > > > ### Author Response · Authors · 2024-12-02
> > > >
> > > > As the discussion period is drawing to a close, we wanted to kindly check if there are any remaining questions or concerns we can address to clarify or strengthen our submission. Thank you again for your time and thoughtful feedback throughout this process!

---

> > > > > ### Author Response · Authors · 2024-12-03
> > > > >
> > > > > We hope our responses have addressed your outstanding questions and clarified any concerns. If so, we would sincerely appreciate it if you could consider revisiting your score. Thank you again for your thoughtful review and valuable feedback—it has been helpful in improving our work!

---

> > > > > > ### Comment · Reviewer_CqcH · 2024-12-03
> > > > > >
> > > > > > Dear Authors,
> > > > > >
> > > > > > I thank you for your replies. I read other reviews and I have to agree with the reviewer  9Vwg that your comparison of neurons in awake and anaesthetised state suffers from an important drawback, which is that the awake and anestetized neural activities are from two distinct datasets, recorded in two different labs. I apologise that I did not notice this before and did not raise this concern.
> > > > > > There can be significant differences in the way recordings were made and in a number of preprocessing steps taken, and these differences seem close to impossible to control or to eliminate from already preprocessed data. Moreover, these differences could easily result in major differences across the two datasets, that would not be due solely to the change in the state from awake to anesthetized. For a more fair comparison, the awake and anestetized neural activities should be recorded in the same set of experiments and following exactly the same procedure for preprocessing. Alternatively, it would be possible and to some degree also insightful, even though less rigorous than the first strategy, to do a meta-analysis that would include several datasets of anesthetized and awake neural activities.
> > > > > >
> > > > > > In light of this concern, I unfortunately had to decrease my score for soundness, and with this also the general score.

---

> > > > > > > ### Author Response · Authors · 2024-12-03
> > > > > > >
> > > > > > > Dear reviewer, thank you for your feedback.
> > > > > > >
> > > > > > > We discussed this shortcoming of our work in our initial submission, and have significantly extended the discussion in our revised submission (see the "Experimental limitations" paragraph). All our control experiments show that the awake neurons are more variable in their tuning compared to the anetheized neurons, as evident when the number of training samples are matched (updated Appendix Fig. 5); and when the number of training samples and the number of neurons are matched (new Appendix Fig. 6). You can see that the DSTRFs in the awake model trained with a matching number of training samples qualitatively also exhibits dynamics tuning (new Appendix Fig. 4). Although the DSTRF activity (the fraction of time the dynamic-STRF is ”on”) becomes more similar, we found their tuning variability to be significantly higher in the awake neurons than the anesthetized neurons when comparing the DSTRFs just on the awake dataset sounds (new Appendix Fig. 7a,b) and just on the anesthetized dataset sounds (new Appendix Fig. 7c, d).
> > > > > > >
> > > > > > > We hope you may be open to re-raising your score, in light of our extensive reviews addressing this issue.

---

### Official Review · Reviewer_ScyS · 2024-11-04

**Soundness:** 3
**Presentation:** 3
**Contribution:** 2
**Rating:** 6
**Confidence:** 3

**Summary:**

The authors seek to understand the response properties of a "conscious" neural network (e.g., a model that is trained to predict the firing rate of biological auditory cortex neurons during an "awake" state) and a "unconscious" neural network (e.g., a model that is trained to predict the firing rate of biological auditory cortex neurons during an "anesthetized" state). Interestingly, augmenting the encoding model with a "gating" mechanism improves the firing rate predictions.

**Strengths:**

The paper and results are very interesting, and has potential impact to both neuroscience and machine learning. In particular, the formal connection between the neural processing of auditory stimuli and the effect of anesthetics is inspiring. The paper is well-written and the formulation of the encoding model(s) and training setup is clear.

The result on the transition from "polysemanticity" of the "awake" A1 neurons to "monosemanticity" of the "anesthetized" neurons is quite interesting.

**Weaknesses:**

* There is no formal comparison of the "semanticity" of the biological A1 neurons and the artificial neurons in the (trained) encoder models (e.g., TC model). For example, are the "awake" TC neurons more polysemantic, while the  "anesthetized" TC neurons more monosemantic?

* The role of "gating" in these models, albeit briefly motivated, is quite sparse. For instance, how does the proposed "gating" compare to "dynamic thresholding," which aligns more closely with the observed properties of cortical sensory neurons. Dynamic thresholds allow neurons to activate only when inputs meet a certain criterion, essentially filtering out less relevant signals without directly scaling the output.

**Questions:**

* In Figure 1, the authors bin the spike trains into $4$ ms intervals, leading to very high (possibly non-biological) firing rates ($150$-$250$ Hz). Do the A1 neurons in the Ferret actually display this range of firing rates?

* In sequence models(e.g., LSTM), the "gating" mechanism controls the routing of information to the next cell. In this work, what is the role of the gating mechanism in relation to anesthesia? In an anesthetized state, there is no conscious perception, so information of certain sounds doesn't reach higher-level brain regions, such as the hippocampus. In effect, there is more "gating" of information in an anesthetized state. However, in Figure 5, the results suggest the "awake" state displays more "gating" (e.g., more incoming connections to each output unit) than the "anesthetized" state, which displays less "gating."

* Certain anesthetics are known to be amnesic (induce forgetting). In an LSTM, the "forget" gate removes irrelevant information. Does the "gating" in the anesthetized model learn to discard irrelevant information?

---

> ### Author Response · Authors · 2024-11-26
>
> We thank the reviewer for their positive assessment, and we hope that our answers below satisfy their comments.
>
> **W1: There is no formal comparison of the "semanticity" of the biological A1 neurons and the artificial neurons in the (trained) encoder models (e.g., TC model). For example, are the "awake" TC neurons more polysemantic, while the "anesthetized" TC neurons more monosemantic?**
>
> > Apologies for the confusion. You are correct in your assessment. We infer the “semanticity” of the biological A1 neurons from the predicted responses of the gated-TC model. This is done by 1. Fitting the model to the neuron responses 2. Obtaining the DSTRFs for each neuron using the model’s predicted responses 3. Calculating the tuning variability from the DSTRFs. We have now made this more explicit in section 3.3.
>
> **W2: The role of "gating" in these models, albeit briefly motivated, is quite sparse. For instance, how does the proposed "gating" compare to "dynamic thresholding," which aligns more closely with the observed properties of cortical sensory neurons. Dynamic thresholds allow neurons to activate only when inputs meet a certain criterion, essentially filtering out less relevant signals without directly scaling the output.**
>
> > You raise an interesting question. We have now further investigated the role of the gating mechanism in the TC model and found it to reduce the prediction error when the neuron responses are particularly elevated (see updated Fig. 1c). Thus, it appears that gating improves the response prediction by scaling the output responses during time periods when the neuron responses are high.
>
> > We imagine that your idea of dynamic thresholding may perhaps improve the response predictions when the activity is particularly low. Thus, joining these two mechanisms of gating and thresholding may further improve the response predictions of the models. We have now elaborated on this in our Discussion (perhaps there is relevant work you would like us to cite) and we look forward to exploring this idea in future work.
>
> **Q1: In Figure 1, the authors bin the spike trains into 4 ms intervals, leading to very high (possibly non-biological) firing rates (150-250 Hz). Do the A1 neurons in the Ferret actually display this range of firing rates?**
>
> > Yes, ferret A1 neurons have previously been reported to spike within this range, see Harper et al. (2016). In theory, neurons could spike at higher rates (500Hz) as their spike rates are bound by their absolute refractory period which is usually around ~2ms (Rolls, 1971). However, we found the mean firing rate to  generally be low across all neurons on both the awake (14.28Hz) and anesthetized (12.54Hz) datasets (see new Appendix Fig. 2). We have now mentioned this in the “A1 neural datasets” paragraph in our Methods.
>
> > Harper, N.S., Schoppe, O., Willmore, B.D., Cui, Z., Schnupp, J.W. and King, A.J., 2016. Network receptive field modeling reveals extensive integration and multi-feature selectivity in auditory cortical neurons. PLoS computational biology, 12(11), p.e1005113.
>
> > Rolls, E.T., 1971. Absolute refractory period of neurons involved in MFB self-stimulation. Physiology & Behavior, 7(3), pp.311-315.
>
> **Q2: In sequence models(e.g., LSTM), the "gating" mechanism controls the routing of information to the next cell. In this work, what is the role of the gating mechanism in relation to anesthesia? In an anesthetized state, there is no conscious perception, so information of certain sounds doesn't reach higher-level brain regions, such as the hippocampus. In effect, there is more "gating" of information in an anesthetized state. However, in Figure 5, the results suggest the "awake" state displays more "gating" (e.g., more incoming connections to each output unit) than the "anesthetized" state, which displays less "gating."**
>
> > Apologies for not making this more clear in our paper. In Fig. 5 we showcase the connectivity of the gated-TC model into the output units. The output of the unit is the linear sum of the connectivity values and their inputs passed through a non-linearity, which is then multiplied by the gating term (see Eq. 6). Thus, more incoming connections does not imply more gating and they are separate entities. We have now updated section 3.5 to make this more clear.
>
> > Regarding the role of the gating mechanism in relation to anesthesia: in our new analysis in updated Fig. 1c we found gating to improve the response predictions during periods of time when the neuron responses are particularly elevated. However, this improvement was similar between for the model trained on the awake (3.6%) and anesthetized dataset (4.4%).  Thus, it appears that gating plays a similar role during conscious and unconscious auditory cortical processing. We further describe this in section 3.1.

---

> > ### Author Response · Authors · 2024-11-26
> >
> > **Q3: Certain anesthetics are known to be amnesic (induce forgetting). In an LSTM, the "forget" gate removes irrelevant information. Does the "gating" in the anesthetized model learn to discard irrelevant information?**
> >
> > > Following our last responses, we found the gating mechanism to correctly scale the response predictions during time periods when the neuron activity was particularly elevated. Thus, it appears that the gating mechanism in our explored auditory encoding models rather “amplifies” sensory information, rather than forgetting it like in LSTMs. We have now mentioned this in our Discussion.

---

> > > ### Author Response · Authors · 2024-12-01
> > >
> > > Dear reviewer, we hope our responses have addressed your questions and clarified any concerns. As the extended discussion period comes to an end, please let us know if you have any further questions preventing you from raising your score. We greatly appreciate your time and effort.

---

> > > > ### Author Response · Authors · 2024-12-02
> > > >
> > > > As the discussion period is drawing to a close, we wanted to kindly check if there are any remaining questions or concerns we can address to clarify or strengthen our submission. Thank you again for your time and thoughtful feedback throughout this process!

---

> > > > > ### Comment · Reviewer_ScyS · 2024-12-02
> > > > > **Official Response by Reviewer**
> > > > >
> > > > > After reviewing the updated revisions provided by the authors, I find that they have largely addressed my concerns, and their revisions significantly improve the quality of the work. Consequently, I am raising my score to a 6, as the paper provides interesting empirical insights of auditory processing of neurons in different ``states" (awake, anesthetized) with gating mechanisms, which I believe should be explored further.
> > > > >
> > > > > One limitation, however, is that "awake" TC neurons exhibit more complex, dynamic spectrotemporal tuning, making them harder to interpret than "anesthetized " TC neurons. Future work could investigate approaches to better interpret and characterize this complexity in ``awake" TC neurons.
> > > > >
> > > > > **``Thus, joining these two mechanisms of gating and thresholding may further improve the response predictions of the models."**
> > > > >
> > > > > One *possible* suggestion here is: a two-stage modulation system, where in Stage 1 you have Dynamic Thresholding, acting as a pre-filter at the input stage of each neuron, and then in Stage 2 there is Gating, modulating the output of the neuron by scaling its response based on some contextual factors (e.g., the network state or overall activity level).

---

> > > > > > ### Author Response · Authors · 2024-12-03
> > > > > >
> > > > > > We thank you for your positive feedback and suggestions, and we agree that there are many exciting avenues to further develop the gating mechanism.

---

### Official Review · Reviewer_9Vwg · 2024-11-04

**Soundness:** 1
**Presentation:** 2
**Contribution:** 2
**Rating:** 3
**Confidence:** 5

**Summary:**

This work looks at auditory neurons collected during awake and anesthetized recordings to try and identify differences in the coding properties that differ during conscious and unconscious brain states. The authors use various forms of computational modeling and interpretability tools to investigate how the responses differ. The authors claim that neurons under anesthesia respond to single inputs (a lower dimensional code), while neurons in awake animals respond to multiple inputs (a higher dimensional code).

**Strengths:**

The paper presents a creative method of addressing the question of differences between awake and anesthetized neural responses. The goals of the paper are ambitious. The idea is to combine predictive modeling approaches and neural network interpretability tools with neural data to learn new things about the neural code. In doing so, the authors aim to unify conflicting claims about auditory processing.

**Weaknesses:**

The biggest weakness of the paper is the confounds between the awake and anesthetized datasets (this is noted by the authors in the discussion). The datasets are collected on different sound sets, with significantly more sounds in the awake recordings than in the anesthetized recordings. Even though the authors in one experiment show a control where they match the number of stimuli, there is not enough detail given on how these stimuli were chosen and the differences in sounds between the two datasets. Significantly more control analyses would need to be run to make sure that the differences observed are not simply due to differences between the datasets. Unfortunately, the best way to test this hypothesis would be for data to be collected in the same animal during awake and anesthetized recordings, which is far beyond the scope of an ICLR rebuttal period.

Overall though, I found the paper to be using somewhat overly complicated methods to answer fairly standard questions, and description of the methods and results was not the easiest to follow (many critical details are missing, see some examples in the questions). I found myself wondering whether the methods used were actually necessary to ask the question of whether the responses under anesthesia become lower dimensional -- the PCA analysis by itself might be sufficient in a controlled neural dataset.

**Questions:**

1) Could the authors provide more detail about the two different sound datasets? How many repetitions were presented for each sound? What was the general composition of sound events of each dataset?

2) Could the differences between results for the two datasets be explained by the recording locations of the neurons, or perhaps spike-sorting/preprocessing differences (and if not, what evidence is there that this isn’t the case)? The two datasets were collected by different labs and presumably very different recording setups.

3) Is the L2 regularization constraint for feature visualization (line 134) applied on the cochleagram? Why is this a reasonable constraint to use for the cochleagram (what is being “smoothed”?)

4) On line 120 it is stated that a batch size of 64 sound clips is used for training, but one of the datasets only had 20 sound clips. Is there a different training dataset used in this section?

5) I had some general questions about the PCA analysis. In figure 3 and the explanation of these results (lines 299-305) it would be helpful to remind readers what PCA is being performed on. As it currently reads, it seems like it is being done on something within UMAP (which would be a problem), but I believe it is on the raw DSTRF responses (the correct thing to do). Also, for this PCA in Figure 3 and also the PCA in Figure 4, is there any preprocessing done to the signal before PCA is performed? And what does each PCA analysis look like when using a matched number of stimuli and neurons (ie can the differences in this section be explained by the dataset differences?)

6) For Figure 4d, it is noted that the number of neurons was subsampled so the number was matched between two datasets, but what if the number of neurons *and* the number of sounds is matched between the two datasets?

7) It might be helpful to have a beginning of the discussion section that gives the overall summary of the results, and how all of the results fit together.

---

> ### Author Response · Authors · 2024-11-26
>
> We thank you for your constructive criticism. We agree that the confound between the awake and anesthetized datasets is a shortcoming. We have now run more extensive analysis and control experiments to ensure that our reported results are less likely to arise from the confounding factors between the datasets, which we outline in our responses to your questions below.  Furthermore, we have provided more details regarding our analysis throughout the paper.
> We also agree that some of our analysis may be more than required to answer our question of investigation (e.g. using the SAEs in addition to the PCA to inspect the dimensionality of the population codes). However, this is not a shortcoming of our work, but rather a strength in that it further corroborates our findings, where we show that different methods point to a similar outcome.
> We hope our extensive revisions address your concerns. Please let us know if you have any more concerns or questions, which we will happily address.
>
>
> **Q1: Could the authors provide more detail about the two different sound datasets? How many repetitions were presented for each sound? What was the general composition of sound events of each dataset?**
>
> > We apologize for not making this more clear in our paper. The anesthetized dataset has 20 repeats for each sound in the training and test set, and the awake dataset has no repeats for the training set and 10 repeats for the test set. We trained the encoding models to predict the trial-averaged responses to the sounds. Both datasets include sounds of human speech, animal vocalizations, and environmental sounds. We have now made this clearer in our Methods section. In addition, we now include a new Appendix Fig. 1 showcasing the average power spectrum of the natural sound stimuli of both datasets. Both datasets showcase a similar declining power profile, with the highest power in the lower frequencies and the lowest power in the higher frequencies, as has similarly been reported in prior work (Machens et al. 2004).
>
> > Machens, C.K., Wehr, M.S. and Zador, A.M., 2004. Linearity of cortical receptive fields measured with natural sounds. Journal of Neuroscience, 24(5), pp.1089-1100.
>
> **Q2: Could the differences between results for the two datasets be explained by the recording locations of the neurons, or perhaps spike-sorting/preprocessing differences (and if not, what evidence is there that this isn’t the case)? The two datasets were collected by different labs and presumably very different recording setups.**
>
> > Both datasets consist of extracellular recordings of neurons in ferret A1 (and some neurons from the anterior auditory field) spanning the tonotopic gradient. Therefore, both datasets should contain recordings from A1 neurons with similar frequency tuning properties. Indeed, we found the averaged population STRF of the neurons in the awake and anesthetized datasets to exhibit a temporally asymmetric power profile with excitation near the present followed by lagging inhibition into the past (new Appendix Fig. 9), as has similarly been reported in prior work (deCharms et al., 1998).
>
> > Although the datasets were recorded using different hardware and spike-sorting software, they have previously been used in prior work for side-by-side comparison, where Rahman et al. (2020) used these datasets to contrast the performance of various cochleagram models in predicting auditory cortical responses under awake and anesthetized conditions. Although the anesthetized recordings were done using silicon probe electrodes and the awake recordings were done using tungsten microelectrodes, these systems have previously been shown to provide comparable action potential recordings (Saha et al. 2010). Furthermore, both datasets were gathered from reputable labs, which have likely calibrated their recording systems to be as precise as possible. Thus, it seems unlikely that the differences between the different labs' experimental setups would explain the auditory tuning differences we report. We have now further expanded upon this in our Methods and limitations paragraph in the Discussion.
>
> > DeCharms, R.C., Blake, D.T. and Merzenich, M.M., 1998. Optimizing sound f	eatures for cortical neurons. science, 280(5368), pp.1439-1444.
>
> > Rahman, M., Willmore, B.D., King, A.J. and Harper, N.S., 2020. Simple transformations capture auditory input to cortex. Proceedings of the National Academy of Sciences, 117(45), pp.28442-28451.
>
> > Saha, R., Jackson, N., Patel, C. and Muthuswamy, J., 2010. Highly doped polycrystalline silicon microelectrodes reduce noise in neuronal recordings in vivo. IEEE Transactions on Neural Systems and Rehabilitation Engineering, 18(5), pp.489-497.

---

> > ### Author Response · Authors · 2024-11-26
> >
> > **Q3: Is the L2 regularization constraint for feature visualization (line 134) applied on the cochleagram? Why is this a reasonable constraint to use for the cochleagram (what is being “smoothed”?)**
> >
> > > The L2 regularization for the feature visualization is applied to the input weights which are trained via gradient descent to maximally activate a particular model unit response. After optimization, these input weights give an estimate of a units spectrotemporal tuning preference. No sound cochleagram is used during this procedure. It is common to include a smoothing regularization term for feature visualization to promote smoothness within the learnt receptive fields (Olah et al. 2017). We have now made this more explicit in section 3.2. We include new Appendix Fig. 3 to showcase what the spectrotemporal receptive fields obtained using feature visualization look like when employing different levels of L2 regularization, where lower L2 regularization values result in receptive fields with higher power across frequencies and time.
> >
> > > Olah, C., Mordvintsev, A. and Schubert, L., 2017. Feature visualization. Distill, 2(11), p.e7.
> >
> > **Q4: On line 120 it is stated that a batch size of 64 sound clips is used for training, but one of the datasets only had 20 sound clips. Is there a different training dataset used in this section?**
> >
> > > We apologize for the confusion. We used up to 64 sound clips as the training batch size. However, as the anesthetized dataset contains less sound clips, this meant we trained in full-batch mode. We have now made this more clear in the Methods.
> >
> > **Q5: I had some general questions about the PCA analysis. In figure 3 and the explanation of these results (lines 299-305) it would be helpful to remind readers what PCA is being performed on. As it currently reads, it seems like it is being done on something within UMAP (which would be a problem), but I believe it is on the raw DSTRF responses (the correct thing to do). Also, for this PCA in Figure 3 and also the PCA in Figure 4, is there any preprocessing done to the signal before PCA is performed? And what does each PCA analysis look like when using a matched number of stimuli and neurons (ie can the differences in this section be explained by the dataset differences?)**
> >
> > > We agree that the text reads somewhat confusing and apologize for the ambiguity. The PCA was performed on the DSTRFs and not on the UMAP embeddings. There is no preprocessing done before performing PCA. We have now made this more explicit in the section 3.3.
> >
> > > In the PCA analysis in section 3.3, we performed PCA separately on each DSTRF and not collectively over all the neuron DSTRFs. The “tuning variability” measure relates the spread of explained variance over the principal components of each distinct DSTRF into a single number, where a lower number implies a more fixed spectrotemporal structure and a higher number implies a more dynamic spectrotemporal structure for a given DSTRF. In our original analysis, we trained the gated-TC encoding model on both datasets with a matching number of stimuli (original Appendix Figure). Here, we uniformly sampled 16 training stimuli from the awake dataset. We have now re-run this analysis of training the model and calculating the DSTRFs five different times, each time using a different random subsample of sounds from the awake dataset. We found the DSTRFs to still qualitatively exhibit dynamic tuning (see side-by-side comparison in new Appendix Fig. 4); and again, we found the tuning variability to be significantly higher in the awake neurons than the anesthetized neurons (updated Appendix Fig. 5). Further matching the number of neurons between the datasets by uniformly subsampling 73 random neurons from the awake dataset for each of the stimulus-matched experiments does not alter the significance (new Appendix Fig. 6).
> >
> > > We have additionally run more thorough analysis for comparing the tuning properties between the awake and anesthetized datasets by comparing the DSTRFs on the same sound datasets (and not between the different sound datasets). This is possible as the DSTRFs can be calculated for any given cochleagram input. Again, we found the tuning variability to be significantly higher in the awake neurons than the anesthetized neurons when comparing the DSTRFs just on the awake dataset sounds (new Appendix Fig. 7a) and just on the anesthetized dataset sounds (new Appendix Fig. 7c). These results still hold true when matching the number of training stimuli (new Appendix Fig. 7b and d).

---

> > > ### Author Response · Authors · 2024-11-26
> > >
> > > **Q6: For Figure 4d, it is noted that the number of neurons was subsampled so the number was matched between two datasets, but what if the number of neurons and the number of sounds is matched between the two datasets?**
> > >
> > > > We calculated the results in Fig. 4 using the held-out test datasets which contain an equal number of sounds. Thus, both the number of neurons and the number of sounds were matched in this analysis. We have now made this more explicit in the section 3.4.
> > >
> > > >To further study the dimensionality of the population code using more stimuli, we have re-run the PCA analysis reported in Fig. 4b using the training datasets (instead of the held-out test dataset), and uniformly subsampled the number of neurons and sounds in the awake dataset to match the anesthetized dataset. We repeated this random subsampling five different times, each time with a different random seed. Again, we observed the awake population responses to exhibit a higher dimensional code than the anesthetized population responses (new Appendix Fig. 8).
> > >
> > > **Q7: It might be helpful to have a beginning of the discussion section that gives the overall summary of the results, and how all of the results fit together.**
> > >
> > > > We agree and have added the following beginning paragraph to our Discussion: “We explored how general anesthesia affects sensory processing in the primary auditory cortex (A1) using auditory encoding models trained to predict neuronal responses to natural sounds in awake and anesthetized ferrets. Utilizing mechanistic interpretability techniques, we discovered that anesthesia induces a shift in neuronal tuning from polysemantic - where neurons respond to multiple spectrotemporal features - to monosemantic, responding to a single spectrotemporal input. Furthermore, we found this transition to reduce the dimensionality of the population code due to a decoupling of connectivity in the models.”

---

> > > > ### Comment · Reviewer_9Vwg · 2024-11-26
> > > > **quick note after author response**
> > > >
> > > > Thank you for posting an updated version of the paper and comments! It looks like there are quite a few changes, and I want to make sure to spend enough time on them. Since the discussion period has been extended, I'll send some comments in the next few days.

---

> > > > > ### Author Response · Authors · 2024-12-01
> > > > >
> > > > > Dear reviewer, we hope our responses have addressed your questions and clarified any concerns. As the extended discussion period comes to an end, please let us know if you have any further questions preventing you from raising your score. We greatly appreciate your time and effort.

---

> > > > > > ### Comment · Reviewer_9Vwg · 2024-12-02
> > > > > > **reviewer response to author rebuttal**
> > > > > >
> > > > > > I appreciate the authors’ thoughtful responses and additional experiments. Upon looking through the responses and the paper, I decided to maintain my scores for ICLR 2025. The authors’ responses unfortunately do not fully address my concern about the confounds between the two datasets. Additionally, in light of the extensive revisions that have been made to the paper, I believe that the modified paper requires a full round of review. A few notes below about the author responses to hopefully further improve the paper.
> > > > > >
> > > > > > Re: Appendix Fig 1: Why is the average power spectrum the thing to look at here for whether the datasets are matched? In general, a 1/f like decay will be present in many natural sound datasets, so this doesn’t address my concern about the sound datasets being quite different in number of sounds, number of repetitions, and diversity of recordings.
> > > > > >
> > > > > > Re: L2 regularization: For feature visualization, the “input weights” here need to correspond to the signal that is being optimized (and should be described as such, rather than “weights”). Is this the audio waveform? Is it the cochleagram? In either case, the linked reference for feature visualization applies for *images* and not audio, where this regularization has been motivated as a natural image prior (and even then, it has been debated whether this type of prior is a reasonable assumption for meaningful feature visualization). I know of no such reference motivating this type of prior for *audio* signals.
> > > > > >
> > > > > > Re: Confounds: Thank you for the additional experiments in the appendix. Some of these should replace the analyses performed in the main text to eliminate the dataset confounds, as this awake/anesthetized comparison is the *main* experiment and claim of the paper (this is in contrast to the referenced paper by Rahman et al. which analyzes both datasets, but it is not the primary focus of the paper).

---

> > > > > > > ### Author Response · Authors · 2024-12-02
> > > > > > >
> > > > > > > Thank you for carefully reviewing our responses and for providing detailed feedback. Could you clarify if there are specific aspects of our analyses that did not fully address your concerns?
> > > > > > >
> > > > > > > **Re: Appendix Fig 1: Why is the average power spectrum the thing to look at here for whether the datasets are matched? In general, a 1/f like decay will be present in many natural sound datasets, so this doesn’t address my concern about the sound datasets being quite different in number of sounds, number of repetitions, and diversity of recordings.**
> > > > > > > >We agree that a 1/f decay is a common characteristic in natural sound datasets and acknowledge that the average power spectrum alone may not fully address the dataset matching concerns. However, our intent with this analysis was to highlight that the spectral content of the datasets aligns broadly.
> > > > > > >
> > > > > > > **Re: L2 regularization: For feature visualization, the “input weights” here need to correspond to the signal that is being optimized (and should be described as such, rather than “weights”). Is this the audio waveform? Is it the cochleagram? In either case, the linked reference for feature visualization applies for images and not audio, where this regularization has been motivated as a natural image prior (and even then, it has been debated whether this type of prior is a reasonable assumption for meaningful feature visualization). I know of no such reference motivating this type of prior for audio signals.**
> > > > > > > >Thank you for pointing out the ambiguity in our terminology. We have updated the manuscript to clarify that the input signal being optimized is the cochleagram. While the regularization technique originates from image feature visualization, we believe it is a general approach that can be adapted to any neural network input, including audio signals. That said, we recognize the lack of established references motivating this prior specifically for audio signals and acknowledge this limitation in the revised manuscript.
> > > > > > >
> > > > > > > **Re: Confounds: Thank you for the additional experiments in the appendix. Some of these should replace the analyses performed in the main text to eliminate the dataset confounds, as this awake/anesthetized comparison is the main experiment and claim of the paper (this is in contrast to the referenced paper by Rahman et al. which analyzes both datasets, but it is not the primary focus of the paper).**
> > > > > > > > Thank you for your suggestion. We are open to replacing any specific figures or analyses that you believe would better address the dataset confounds. Could you clarify which figures or results from the appendix you recommend moving to the main text? Additionally, we would like to emphasize that our paper also demonstrates how the proposed gating mechanism improves response prediction across various auditory encoding models, which complements the primary awake/anesthetized comparison.

---

> > > > > > > > ### Author Response · Authors · 2024-12-02
> > > > > > > >
> > > > > > > > As the discussion period is drawing to a close, we wanted to kindly check if there are any remaining questions or concerns we can address to clarify or strengthen our submission. Thank you again for your time and thoughtful feedback throughout this process!

---

> > > > > > > > > ### Comment · Reviewer_9Vwg · 2024-12-03
> > > > > > > > >
> > > > > > > > > Thank you for clarifying that the regularization is indeed applied to the cochleagram representation. However, I fully disagree that it is a "general approach" for any neural network. Any type of regularization imposes a specific prior on the generated signal, which you showed in your supplement, and state in the text: "different levels of regularization can drastically influence the structure of the learnt input weights."  For instance, in Figure 2, certain regions have a higher correlation for anesthetized, and certain regions have a higher correlation for the awake. This shows that it is critical to choose a regularizer that actually yields something like a better fit to held-out data, rather than qualitatively improving visualizations.
> > > > > > > > >
> > > > > > > > > Re: specific examples, I believe that all of the experiments in the main text where a model is fit should be matched to (at minimum) the number of example sounds, number of repetitions of the sound that were used, the number of analyzed units for any multi-unit analysis. It is impossible to evaluate whether changes in things like dimensionality and the different model fits are due to more noise in the awake dataset (due to fewer repetitions per sound) or due to less diversity in the anesthetized data (due to fewer base audio clips). And so Figures 4 and 5 would also need to be updated or presented with the controlled comparison.
> > > > > > > > >
> > > > > > > > > To repeat what I said before, I do think that the paper is interesting and gave this careful consideration, however **I think that the changes are too substantial for this timeline and require a full round of review before publication**. This is especially true given that the control experiments seem to substantially reduce the tuning variability of the awake recordings--the authors even state that "the activity between the awake and anesthetized A1 neurons became more similar" (line 356).

---

> > > > > > > > > > ### Author Response · Authors · 2024-12-03
> > > > > > > > > >
> > > > > > > > > > **Thank you for clarifying that the regularization is indeed applied to the cochleagram representation. However, I fully disagree that it is a "general approach" for any neural network. Any type of regularization imposes a specific prior on the generated signal, which you showed in your supplement, and state in the text: "different levels of regularization can drastically influence the structure of the learnt input weights." For instance, in Figure 2, certain regions have a higher correlation for anesthetized, and certain regions have a higher correlation for the awake. This shows that it is critical to choose a regularizer that actually yields something like a better fit to held-out data, rather than qualitatively improving visualizations.**
> > > > > > > > > > >Feature visualization is a qualitative method for inspecting neural network tuning. There are currently no well-established metrics for evaluating its fit to held-out data. To address this limitation, we selected the regularization value that best matched the correlation to the STRFs for the plots in Fig. 2b - we have now clarified this in the text. We agree with your point that different regularizers impose specific priors, which can influence the results. Exploring these effects further is indeed an open and important research question (Olah et al. 2017) that we aim to investigate in future work. The main goal of Fig. 2 was to illustrate two key findings: (1) For awake neurons, feature visualization reveals spectrotemporal tuning that is less evident in standard STRFs; and (2) For anesthetized neurons, feature visualization produces results that are more similar to those obtained using standard STRFs. We hope this distinction clarifies the intent of the figure and the interpretation of the results.
> > > > > > > > > >
> > > > > > > > > > **Re: specific examples, I believe that all of the experiments in the main text where a model is fit should be matched to (at minimum) the number of example sounds, number of repetitions of the sound that were used, the number of analyzed units for any multi-unit analysis. It is impossible to evaluate whether changes in things like dimensionality and the different model fits are due to more noise in the awake dataset (due to fewer repetitions per sound) or due to less diversity in the anesthetized data (due to fewer base audio clips). And so Figures 4 and 5 would also need to be updated or presented with the controlled comparison.**
> > > > > > > > > > > We agree with your assessment and will update the main text figures to use the awake model trained on an equal number of training samples as the anesthetized model.
> > > > > > > > > >
> > > > > > > > > > **To repeat what I said before, I do think that the paper is interesting and gave this careful consideration, however I think that the changes are too substantial for this timeline and require a full round of review before publication. This is especially true given that the control experiments seem to substantially reduce the tuning variability of the awake recordings--the authors even state that "the activity between the awake and anesthetized A1 neurons became more similar" (line 356).**
> > > > > > > > > >
> > > > > > > > > > > Thank you for your positive feedback of our work. **All our control experiments show that the awake neurons are more variable in their tuning compared to the anetheized neurons**, as evident when the number of training samples are matched (updated Appendix Fig. 5); and when the number of training samples and the number of neurons are matched (new Appendix Fig. 6). You can see that the DSTRFs in the awake model trained with a matching number of training samples qualitatively also exhibits dynamics tuning (new Appendix Fig. 4). Although the DSTRF activity (the fraction of time the dynamic-STRF is ”on”) becomes more similar, we found their tuning variability to be significantly higher in the awake neurons than the anesthetized neurons when comparing the DSTRFs just on the awake dataset sounds (new Appendix Fig. 7a,b) and just on the anesthetized dataset sounds (new Appendix Fig. 7c, d).
> > > > > > > > > >
> > > > > > > > > > We carefully considered all your feedback and made substantial revisions to address your concerns. While we understand that you feel these changes might require a full round of review, we believe the updated manuscript now comprehensively and rigorously addresses your key points. Thanks to your thoughtful input, the revised manuscript is in a much stronger position for publication. Thank you again for your constructive feedback and for recognizing the potential of our work.

---

### Author Response · Authors · 2024-11-26

Dear reviewers,

Thank you all for taking the time to review our work and providing thoughtful feedback. We apologize for the delay in our response. We have tried to address your concerns through additional analyses and clarifications, which are reflected in the revised manuscript (changes in red).

During the revision process, we identified a slight issue with the normalization of the awake dataset. This has now been corrected, and as a result, some of the figures have been slightly updated. Importantly, we found that the tuning properties remained consistent, and the performance of the gated-TC model improved marginally, with CCnorm increasing from 0.63 to 0.64.

To illustrate the differences in tuning, we have included video clips of example DSTRFs for awake and anesthetized neurons. These clips are available for download in the supplementary materials.

---

### Meta-Review · Area_Chair_HDiF · 2024-12-07

**Metareview:**

The primary goal of this work is to use computational modeling to identify differences in sensory processing between conscious and unconscious conditions. The primary finding is that the dimensionality of the neural activity changes and reduces in the unconscious state.

There were a number of concerns for this work that seem to be beyond the ability for a one week back-and-forth to fully address. There are potential confounds in the datasets, as well as questions about the appropriate level of computational modeling needed to achieve the end results. There was significant back and forth, but there is only limited time in this process. Given the subject of the work and the primary findings, perhaps a scientific journal would be a more appropriate venue that will allow a more thorough and complete dialog between the authors and reviewers to properly address these concerns.

**Additional Comments On Reviewer Discussion:**

There were a number of reviewer-prompted revisions. Quite extensive revisions at that, which is in part why it might be more appropriate to have a more time-friendly journal review process, or at the least significant editing for a future submission.

---

### Decision · Program_Chairs · 2025-01-22

Reject